# Evaluation of TILI-2 as an Anti-Tyrosinase, Anti-Oxidative Agent and Its Role in Preventing Melanogenesis Using a Proteomics Approach

**DOI:** 10.3390/molecules27103228

**Published:** 2022-05-18

**Authors:** Anupong Joompang, Preeyanan Anwised, Sompong Klaynongsruang, Sittiruk Roytrakul, Lapatrada Taemaitree, Nisachon Jangpromma

**Affiliations:** 1Department of Biochemistry, Faculty of Science, Khon Kaen University, 123 Mittraphap Road, Muang District, Khon Kaen 40002, Thailand; anupong.jo@go.buu.ac.th (A.J.); somkly@kku.ac.th (S.K.); 2Protein and Proteomics Research Center for Commercial and Industrial Purposes (ProCCI), Faculty of Science, Khon Kaen University, 123 Mittraphap Road, Muang District, Khon Kaen 40002, Thailand; preeyanan.biochem@gmail.com; 3Functional Ingredients and Food Innovation Research Group, National Center for Genetic Engineering and Biotechnology, National Science and Technology Development Agency, Khlong Luang, Pathum Thani 12120, Thailand; sittiruk@biotec.or.th; 4Department of Integrated Science, Faculty of Science, Khon Kaen University, 123 Mittraphap Road, Muang District, Khon Kaen 40002, Thailand; lapata@kku.ac.th

**Keywords:** peptide, monophenolase activity, tyrosinase inhibitor, antioxidant, mass spectrometry

## Abstract

There is a desire to develop new molecules that can combat hyperpigmentation. To this end, the N-terminal cysteine-containing heptapeptide TILI-2 has shown promising preliminary results. In this work, the mechanism by which it works was evaluated using a series of biochemical assays focusing on known biochemical pathways, followed by LC-MS/MS proteomics to discover pathways that have not been considered before. We demonstrate that TILI-2 is a competitive inhibitor of tyrosinase’s monophenolase activity and it could potentially scavenge ABTS and DPPH radicals. It has a very low cytotoxicity up to 1400 µM against human fibroblast NFDH cells and macrophage-like RAW 264.7 cells. Our proteomics study revealed that another putative mechanism by which TILI-2 may reduce melanin production involves the disruption of the TGF-β signaling pathway in mouse B16F1 cells. This result suggests that TILI-2 has potential scope to be used as a depigmenting agent.

## 1. Introduction

The overproduction and accumulation of melanin can result in dermatological disorders such as melasma, freckles, age spots, and other hyperpigmentation effects [1,2,3]. The melanogenesis stimuli [4] can be either extrinsic, such as UVB, or intrinsic, such as hormone and aging, and typically result in an increase of reactive oxygen species (ROS) and reactive nitrogen species (RNS) that lead to the up-regulation of melanogenic transcription factors and proteins such as tyrosinase, TRP-1, and TRP-2 [4,5]. Among the melanogenic proteins, tyrosinase (monophenol monooxygenases, EC 1.14.18.1) is a rate-limiting enzyme that catalyzes two essential reactions in melanogenesis. If the substrate is L-tyrosinase, hydroxylation and subsequent oxidation of L-tyrosine occurs to give L-dopaquinone. If the substrate is L-DOPA, oxidation occurs to give L-dopaquinone. L-dopaquinone–a branch point of eumelanogenesis or pheomelanogenesis–is further converted via numerous steps to finally become eumelanin and pheomelanin. The hydroxylation of L-tyrosine is called the monophenolase activity, while the oxidation of L-DOPA is well-known as the diphenolase activity [6,7]. Consequently, to prevent accumulation of melanin, there is a significant desire to find depigmenting agents that inhibit tyrosinase monophenolase or diphenolase activity [8,9], oxidative stress [10,11], and other related cell signaling pathways [12].

Bioactive peptides are a promising group of depigmenting agents because they are derived from natural sources and therefore display low cytotoxicity. For example, the peptides produced by hydrolysis of the jellyfish proteins have displayed good antioxidative and tyrosinase inhibitory activity [13], while purified jellyfish collagen peptide (JCP) has shown melanogenesis-inhibitory activity [14,15]. CT-2 (LQPSHY), a C-terminus tyrosine peptide derived from rice bran, even repressed melanogenesis in mouse B16 melanomas [16].

Previously, we reported Tyrosinase Inhibitor Leucrocin I-2 (TILI-2), a peptide derivative from Leucrocin I, can inhibit tyrosinase diphenolase activity and could potentially decrease melanin in cells [17]. This work builds upon these preliminary findings. Herein, we look at the ability of TILI-2 to alter tyrosinase monophenolase inhibitory, its free radical scavenging activity, and its wider cellular cytotoxicity against macrophages and human fibroblasts, before using LC-MS/MS proteomics to reveal its implications on the cellular processes in a model mouse melanin cell line B16F1. Our results suggest that TILI-2 has good potential as a depigmenting agent and lays the foundation for further investigation of similar peptides.

## 2. Results and Discussion

### 2.1. Monophenolase Inhibitory Activity

In melanin synthesis, tyrosinase is a rate-limiting enzyme that displays both monophenolase and diphenolase activities [18]. As a monophenolase, tyrosinase hydroxylates L-tyrosine. As a diphenolase, tyrosinase oxidizes L-DOPA [6,7]. Previously, TILI-2 was reported to inhibit diphenolase with IC_50_ of 113.53 ± 3.76 µM [17]. However, the monophenolase inhibitory activity of this peptide was not evaluated. Here, we demonstrate that it can inhibit monophenolase activity in a dose-dependent manner (12.98–52.68% for 0–800 µM TILI-2 relative to 31.71–53.76% for 0–200 µM kojic acid, a positive control). The IC_50_ of TILI-2 was 582.92 ± 6.44 µM (Figure 1 and Table 1), while the IC_50_ of known monophenolase inhibitor kojic acid was 173.20 ± 6.44 µM (Figure 1 and Table 1). This suggests that TILI-2 is a relatively potent monophenolase inhibitor. The multi-modal action of TILI-2 against tyrosinase is promising and implies it could be a more potent inhibitor peptide. 

### 2.2. Kinetic Study

Next, a kinetic study was performed in order to evaluate the type of inhibition TILI-2 causes. A Lineweaver-Burk plot revealed that TILI-2 increased the Km without affecting the Vmax (Figure 2 and Table 2), suggesting it is a competitive inhibitor. This is consistent with the previous observations made for its mode of diphenolase inhibition [17]. In addition, re-plotting of the slope vs TILI-2 concentration enabled determination of the K_i_ value was 273.39 ± 72.90 µM. As a consequence of these results, we hypothesize that TILI-2 competes with L-tyrosine to bind with the tyrosinase active site. This supports previous findings for similar terminal cysteine containing di-[19], tri-[20], and tetrapeptide [21] that are thought to coordinate to the copper atoms of tyrosinase active site via the thiol groups, resulting in competitive inhibition.

### 2.3. Antioxidant Activity

There is a growing body of evidence that suggests reactive oxygen species (ROS) play a critical role in promoting melanogenesis [22,23], and as a consequence, antioxidant agents have the potential to suppress melanogenesis [10,11]. To this end, peptides are known to have antioxidant activity in addition to their depigmenting ability [24,25]. Their antioxidative capacity is influenced by the peptide size and amino acid composition, with peptides that contain sulfur-containing amino acid residues (Cys and Met) being particularly potent [26]. TILI-2 is short and importantly contains an N-terminal cysteine. As a result, the antioxidant activity of TILI-2 toward ABTS^•+^ and DPPH^•+^ was evaluated (Figure 3). The results showed that TILI-2 could scavenge ABTS and DPPH radicals by 13.95–80.58% and 22.98–78.48% respectively, which is comparable to the effects of the control–ascorbic acid (11.64–82.68% and 13.79–82.63% respectively). The IC_50_ value of TILI-2 for ABTS radical scavenging was determined to be 16.78 ± 0.48 µM, which is similar to that of the ascorbic acid (15.86 ± 0.16 µM; Table 3 and Figure 3) control. In addition, the TILI-2 and ascorbic acid exhibited the IC_50_ value of 89.83 ± 2.72 µM and 40.09 ± 0.21 µM for DPPH radical scavenging, respectively (Table 3). These finding suggests that TILI-2 is a relatively potent antioxidant and could be another mode by which to help in hyperpigmentation treatment.

### 2.4. Cytotoxicity

Next, the cytotoxicity of TILI-2 toward normal human fibroblasts (NFDH) and macrophages (RAW 264.7) was determined in order to evaluate its efficacy. Macrophages are a type of white blood cell [27], which are present in all tissue, including skin [28]. They are very sensitive to most toxic chemicals [29], therefore, the cytotoxicity of TILI-2 toward this cell might provide more clues about the safety of utilizing TILI-2. The results showed that TILI-2 was not toxic against all tested cells. From 0 to 1400 µM, TILI-2 displayed a negligible reduction in cell viability (NHDF: 100 ± 5.70–99.65 ± 1.31%; RAW 264.7: 100 ± 3.20–105.41 ± 5.42%; Figure 4). These results are consistent with previous reports for TILI-2 against human keratinocyte (HaCaT) and mouse melanoma (B16F1) cells [17].

### 2.5. B16F1 Cell Proteomics Profile Based Biological Processes and Cellular Components

The previous assays provided a rational approach to investigating the mechanism by which TILI-2 inhibits melanogenesis. However, compounds rarely have one specific mechanism of action, and in order to improve the effects of TILI-2 in the future, it is desirable to have a more in-depth understanding of the cellular pathways TILI-2 affects. To this end, a LC-MS/MS proteomics study was performed against mouse melanin model B16F1 cells. TILI-2 (350 µM) was administered at the doses that have previously been shown to decrease melanin production and to be non-toxic [17] (Figure 5).

LC-MS/MS spectra were compared with the protein sequences of the Uniprot Database (*Mus musculus*) to identify the expressed proteins. The 3485 proteins were found in both the control and the treated group. Of the 3485 proteins, 6 proteins only were found in the control group, 9 proteins were expressed only in the TILI-2 treated group, and 3470 proteins were detected in both groups (Figure 6).

In order to understand the function of these 3485 proteins, they were categorized by the PANTHER classification system (http://www.pantherdb.org; accessed on 30 January 2021) according to biological processes and the cellular components. The 3485 proteins could be grouped into 22 biological processes, which are mainly assigned in the cellular process (25.10%), metabolic process (15.30%), and biological regulation (14.50%) (Figure 7A). The categorization of protein functions according to the cellular component showed the expressed protein mainly associated in the cell (22.80%), cell parts (22.80%), and organelles (15.60%) (Figure 7B). The obtained proteomics data suggests that TILI-2 affects the overall proteins of the B16F1 cells in various cellular functions.

### 2.6. B16F1 Cell Proteomics Profile Based Cell Signaling Pathway

The 3485 expressed proteins were grouped into 124 cell signaling pathways according to the PANTHER classification system (http://www.pantherdb.org; accessed on 30 January 2021). The signaling pathways that contained a number of proteins greater than or equal to 1% are shown in Figure 8. Among the enriched signaling pathways, transforming growth factor-β (TGF-β) was previously reported to suppress melanogenesis [30,31,32,33,34,35]. TGF-β was found to negatively regulate paired box 3 (PAX3), resulting in the decrease of microphthalmia-associated transcription factor (MITF), whose expression normally elevates melanin synthesis [36]. This suggests that the TGF-β signaling pathway might be critical in decreasing melanin synthesis upon TILI-2 treatment. As a result, the 17 expressed proteins in this pathway (Table 4) were evaluated in more detail.

### 2.7. Protein-Protein Interaction Prediction by STITCH Online Database

The STITCH online database was used to predict the protein–protein interaction between 17 proteins in the TGF-β signaling pathway (Table 4) and well-known proteins in melanogenesis (tyrosinase (TYR), MITF, PAX3, TRP-1 and TRP-2). The results suggest that 7 of 17 proteins are directly involved in melanogenesis, of which MAPK3, MAPK12, MAPK1, Smad5, and BMP4 were predicted to interact with MITF, while TGF-β2 and TGF-β3 were predicted to directly interact with PAX3 (Figure 9).

MAPK3, MAPK12, and MAPK1 are parts of the MAPK family [37]. These proteins specifically phosphorylate the hydroxyl side chains of serine, threonine, and tyrosine residues in their substrates [38]. They play an important role in extracellular transduction, resulting in the cellular responses such as cell proliferation, differentiation, development, inflammation, apoptosis, and stress response [37]. Previously, melanogenesis-inhibiting agents were reported to suppress melanogenesis by activating the MAPK/ERK pathway [39,40,41,42].

Bone morphogenetic protein 4 (BMP4), TGF-β2, and TGF-β3 are signaling ligands belonging to the TGF-β superfamily [34,43]. In melanogenesis, BMP4 was reported to suppress melanogenesis by decreasing tyrosinase expression [44,45]. BMP4 can also induce MITF degradation via the MAPK/ERK pathway resulting in the suppression of melanogenesis [46]. For TGF-β, melanogenesis inhibition is caused by the negative regulation of paired box 3 (PAX3) by the TGF-β/Smad signaling pathway, resulting in the decrease of microphthalmia-associated transcription factor (MITF) [31]. TGF-β3 decreases melanin synthesis by the activation of ERK, resulting in a decrease of MITF and consequently reduced melanogenic enzymes [35].

Smad5 is a part of the R-Smad family, which are the signaling molecules of BMP4 [47]. The binding of BMP4 to its receptor causes the phosphorylation of R-Smad (Smand-1, 5, or 8). These phosphorylated R-Smads then recruit Smad-4, resulting in its translocation into the nucleus, where it interacts with DNA and consequently suppresses or activates the target gene [34]. Smad was reported to be involved with the suppression of PAX3 in the TGF-β/Smad signaling pathway [31].

In summary, the increase of protein expression of TGF-β2, TGF-β3, MAPK3 (ERK1), MAPK12, MAPK1 (ERK2), Smad5, and BMP4, and the decrease of protein expression of PAX3, suggest the suppression of melanogenesis by TILI-2 might involve the TGF-β signaling pathway. 

Finally, although melanin production was reduced in B16F1 cells after TILI-2 treatment, TYRP1 and DCT protein levels were elevated (Table 4). This compensatory mechanism has been observed before when cells are treated with potent tyrosinase inhibitor [48,49,50], whereby tyrosinase mRNA is increased to maintain melanin homeostasis. Therefore, it is possible that the increase of TYP1 and TRP-2 (DCT) expression might be the compensatory mechanism of the B16F1 cell to maintain melanin homeostasis resulting from the melanogenesis inhibitory activity of TILI-2.

### 2.8. Gene Expression Using Real-Time PCR

Based on our proteomic results, TGF-β3, BMP4, and PAX3 are involved in the TGF-β signaling pathway and are strongly implicated in suppression of melanogenesis. Therefore, to corroborate the wide scale proteomics studies, the specific mRNA expression levels of these proteins were evaluated using real-time PCR. The collected results suggest that mRNA expression of BMP4, TGF-β3, and PAX3 increased similarly to the protein expression of the proteomics results (Figure 10). Both results from real-time PCR and proteomics experiments suggest that melanogenesis suppression of TILI-2 might be involved in the TGF-β signaling pathway at both proteome and mRNA levels.

## 3. Materials and Methods

### 3.1. Materials and Reagents

L-tyrosine and mushroom tyrosinase were purchased from Sigma Aldrich (St. Louis, MO, USA). Kojic acid was purchased from TCI (Tokyo, Japan). 2,2′-Azino-bis (3-ethylbenzothiazoline-6-sulfonic acid) diammonium salt (ABTS), DPPH (2, 2-diphenyl-1-picrylhydrazyl), L-Ascorbic acid radical, and dimethyl sulfoxide (DMSO) were purchased from Sigma-Aldrich (St. Louis, MO, USA). Fetal bovine serum (FBS), penicillin/streptomycin, Dulbecco’s modified Eagle’s medium (DMEM) and L-glutamine were purchased from Lonza (Walkersville, MD, USA). Trypsin-EDTA was purchased from Corning Inc. (Manassas, VA, USA). 3-[4, 5-Dimethylthiazol-2-yl]-2, 5-diphenyltetrazolium bromide (MTT) was purchased from Sigma-Aldrich (Eugene, OR, USA).

### 3.2. Peptide Synthesis

The synthesis of TILI-2 was performed by GL Biochem Ltd. (Shanghai, China) using Fmoc solid-phase synthesis. All peptides used had a purity of more than 95% based on mass spectrometry technique.

### 3.3. Assay for Monophenolase Inhibitory Activity

The monophenolase inhibitory assay against monophenolase was performed with slight modifications of the protocol reported by Joompang et al. [17]. Briefly, L-tyrosine (2.4 mM, 50 µL in 50 mM phosphate buffer pH 6.8) was mixed with the peptide (final concentration of 0–800 µM, 47.5 µL) and mushroom tyrosinase (1000 U/mL, 2.5 µL). After mixing all components, the absorbance at 475 nm was continuously monitored at 25 °C using a SpectraMax M5 plate reader (Molecular Devices, Sunnyvale, CA, USA). The monophenolase activity (ΔA475/min) was determined from the linear regression of a plot of A475 nm *vs* time (min). Kojic acid was used as a positive control. The percentage of monophenolase inhibition was then calculated using the equation below: $$\mathrm{Monophenolase}\;\mathrm{inhibition}\;\left(\%\right)=\left(1-\frac{A}{B}\right)\times 100$$
where A and B are the initial rates (∆A475/min) of the enzyme reaction with and without the peptide inhibitor, respectively.

### 3.4. Kinetic Study

A kinetic study was performed by determination of the monophenolase activity in the different concentrations of L-tyrosine (0.10–0.80 mM) with and without different concentrations of TILI-2 (final concentration 0–300 µM) in an otherwise identical protocol to the monophenolase inhibitory activity assay (Section 2.3). V_max_–the rate of the reaction when enzyme is fully saturated–and K_m_–the Michaelis constant or the concentration of substrate at which the enzyme achieves half of its maximum rate of reaction–were obtained from a Lineweaver-Burk plot. These values were used to specify the mode of inhibition. *K*_i_ (inhibition constant) was estimated from the secondary plot of the Lineweaver-Burk plot.

### 3.5. ABTS Radical Scavenging Assay

The ABTS radical scavenging assay was performed with slight modifications of the protocol reported by Re et al. [51]. Briefly, ABTS^+^ was generated by the incubation of 7 mM ABTS with 2.45 mM potassium persulfate for 16 h at room temperature. Next, the ABTS radical solution was diluted until the absorbance at 734 nm was 0.7. The diluted ABTS radical solution (100 µL) was mixed with TILI-2 (final concentration 0–200 µM, 10 µL) and incubated at room temperature for 6 min. After that, the absorbance at 734 nm was measured using a SpectraMax M5 plate reader (Molecular Devices, Sunnyvale, CA, USA). The percentage of ABTS radical inhibition was calculated using the equation below:$$\mathrm{ABTS}\;\mathrm{radical}\;\mathrm{inhibition}\;\left(\%\right)=\left(1-\frac{A734\;\mathrm{sample}}{A734\;\mathrm{control}}\right)\times 100$$
where A734 sample and A734 control are the absorbance at 734 nm of the reaction with and without peptide.

### 3.6. DPPH Radical Scavenging Assay

The DPPH radical scavenging assay was performed with slight modifications of the protocol reported by Jandaruang et al. [52]. Briefly, a DPPH radical solution (0.1 mM, 100 µL) was mixed with TILI-2 (final concentration 0–200 µM, 10 µL) and incubated at room temperature in the dark for 30 min. Then, the reaction was monitored by measuring the absorbance of 515 nm using a SpectraMax M5 plate reader (Molecular Devices, Sunnyvale, CA, USA). DPPH radical inhibition (%) was calculated using the equation below:$$\mathrm{DPPH}\;\mathrm{radical}\;\mathrm{inhibition}\;\left(\%\right)=\left(1-\frac{A515\;\mathrm{sample}}{A515\;\mathrm{control}}\right)\times 100$$
where A515 *sample* and A515 *control* are the absorbance at 515 nm of the reaction with and without peptide.

### 3.7. Cell Culture

Fibroblast cells (NHDF) and B16F1 cells were purchased from the American Type Culture Collection (ATCC, Manassas, VA, USA) and cells were cultured in DMEM supplemented with 10% heat-inactivated FBS, 1% penicillin-streptomycin solution, and 5% L-glutamine. RAW 264.7 cells were purchased from the American Type Culture Collection (ATCC, Manassas, VA, USA) and cells were cultured in RPMI supplemented with 10% heat-inactivated FBS and 1% penicillin-streptomycin solution. These cells were cultured at 37 °C in a 5% CO_2_ and 90% relative humidity incubator.

### 3.8. Cell Viability Assay

Cell viability assay was performed using slight modifications of the protocol described by Phosri et al. [53]. Briefly, fibroblast cells (NHDF) (8000 cells/well) and RAW 264.7 cells (25,000 cells/well) were seeded onto a 96-well plate and allowed to recover overnight. Then, the cells were incubated with different concentrations of peptides (final concentration of 0–1400 µM, 100 µL). After 24 h incubation, the culture medium was removed, and MTT was added (0.5% *w*/*v*, 100 µL). After incubation for a further 30 min at 37 °C, DMSO was added and the absorbance at 570 nm was measured. The cell viability was calculated using the equation below:$$\mathrm{Cell}\;\mathrm{viability}\;\left(\%\right)=\left(\frac{A570\;\mathrm{sample}}{A570\;\mathrm{control}}\right)\times 100$$
where A570 sample and A570 control are the absorbance of cells with and without the peptide treatment.

### 3.9. Melanin Content

The melanin content in cells was measured using the protocol described by Joompang et al. [17], but with slight modifications. B16F1 cells (150,000 cells) were seeded in a 12-well plate and grown overnight. Then, TILI-2 at a final concentration of 350 µM was added. After 48 h, the culture medium was removed, and the cells were washed twice with cold PBS buffer (pH 7.4). The cells were then lysed with cold PBS buffer (0.1 M, pH 7.4) containing 1% Triton X-100 and centrifuged at 12,000 rpm for 30 min. The pellet was collected, washed twice with PBS buffer (0.1 M, pH 7.4), and dried at room temperature. The cell pellet was resuspended with NaOH (aq., 1 M, 110 µL) and incubated at 80 °C for 1 h. The absorbance was measured at 490 nm, and the melanin content was calculated using the equation below:$$\mathrm{Melanin}\;\mathrm{content}\;\left(\%\right)=\left(\frac{A490\;\mathrm{sample}}{A490\;\mathrm{control}}\right)\times 100$$
where A490 sample is the absorbance of cells treated with peptide, and A490 control is the absorbance of cells without peptide.

### 3.10. B16F1 Cell Protein Extraction

Mouse B16F1 cells (150,000 cells/well) were treated with TILI-2 as described in Section 3.9. After the treatment, the cells were harvested using a cell scraper. The cell pellet was collected by centrifuging at 3000 rpm at 4 °C for 5 min. The pellet was washed with ice-cold PBS twice before protein extraction was performed. For protein extraction, the cell pellet was lysed using a trace volume of 50 mM Tris-HCl pH 7.0 containing 0.5% SDS buffer and centrifuged at 10,000 rpm for 15 min [54]. The supernatant was collected, and two volumes of 90% acetone containing 0.07% β-mercaptoethanol solution was added, followed by incubation at −20 °C for 16 h. Then, the mixture was centrifuged at 10,000 rpm for 15 min, and the pellet was resuspended in 10 mM ammonium bicarbonate. The soluble protein concentration was detected using the Lowry method [55]. Bovine serum albumin was used as a standard.

### 3.11. B16F1 Cell Protein Trypsinization and Label-Free LC-MS/MS-Based Proteomics

The protein sample (5 µg, from Section 3.10) was mixed with DTT (5 mM in 10 mM ammonium bicarbonate) and incubated at 60 °C for 1 h in the dark. Iodoacetamide (15 mM in 10 mM ammonium bicarbonate) was then added, and the mixture was incubated at 45 °C for 45 min. Then, the protein sample was mixed with trypsin solution (50 ng trypsin in 50% ACN/10 mM ammonium bicarbonate) at a ratio of 20:1 (*w*/*w*). The digestion reaction was incubated at 37 °C for 12 h. The tryptic peptides were dried at 37 °C before addition of 0.1% formic acid and injection for LC-MS/MS. The peptides were injected into an Ultimate3000 Nano/Capillary LC System (Thermo Scientific, Basingstoke, UK) coupled to a HCTUltra LC-MS system (Bruker Daltonics Ltd.; Hamburg, Germany) equipped with a Nano-captive spray ion source [54,56]. 

### 3.12. Protein Quantitation and Identification

The quantification was carried out by DeCyder MS Differential Analysis software (DeCyderMS, GE Healthcare, Uppsala, Sweden) [57,58]. The identification of proteins was performed using Mascot software (Matrix Science, London, UK) and the NCBI database [59]. The parameters were set as follows; Taxonomy (*Mus musculus*); enzyme (trypsin); variable modifications (carbamidomethyl, oxidation of methionine residues); mass values (monoisotopic); protein mass (unrestricted); peptide mass tolerance (1.2 Da); fragment mass tolerance (± 0.6 Da); peptide charge state (1+, 2+ and 3+); max missed cleavages (1); and instrument (ESI-TRAP). Log2 was used to represent the relative quantitation values. At least one peptide with an individual mascot score at *p* < 0.05 was considered as an identified protein.

### 3.13. Proteomics Data Bioinformatic Analysis

Gene ontology (GO) was evaluated using the PANTHER (Protein Analysis Through Evolutionary Relationships) classification system (http://www.pantherdb.org; accessed on 30 January 2021) to identify the protein function [60]. The STITCH 5.0 database (http://stitch.embl.de/; accessed on 9 February 2021) was used to predict protein–protein interactions [61].

### 3.14. Real-Time PCR

B16F1 cells (150,000 cells/well) were treated with TILI-2 as described in Section 3.9. After treatment, the cells were harvested using a cell scraper. The cell pellet was collected by centrifuging at 3000 rpm at 4 °C for 5 min. The total RNA was extracted from the pellet using the TRIzol^®^ reagent (Invitrogen, CA, USA). Then, total RNA (1 ug) was used to synthesize cDNA using the RevertAid First-Strand cDNA synthesis kit (Fermentas, MA, USA) before real-time PCR was performed using a LightCycler^®^ 480 real-time PCR system (Roche, Rotkreuz, Switzerland). Each real-time PCR reaction (10 µL) consisted of 5 µL of SYBR^®^ Green PCR master mix (Roche, Switzerland), 0.2 μL of forward (10 µM) and reverse (10 µM) primers (Table 5), 1.6 µL of DEPC water, and 3 µL of reverse transcription product. Primer sequences of TGF-β3, BMP4, PAX3, and GAPDH [62,63] are shown in Table 5. GAPDH was used as a housekeeping gene for normalization of target gene expression levels. The measurement of the melting curve was determined to evaluate the fidelity of real-time PCR reaction. The relative change in gene expression was calculated according to Livak and Schmittgen [64]. 

### 3.15. Statistical Analysis

The data were represented as mean ± SD (*n* = 3). An ANOVA test according to Duncan (*p* < 0.05) was used for each comparison (IBM SPSS Statistics 20).

## 4. Conclusions

To summarize, TILI-2 was found to be a competitive inhibitor of tyrosinase monophenolase activity. This supports previous results that it could inhibit tyrosinase’s diphenolase activity [17] and possess a dual mode of action against the enzyme. Further studies demonstrated TILI-2 could potentially scavenge ABTS and DPPH radicals, thereby reducing oxidative stress. Moreover, it has minimal cytotoxicity toward human fibroblast cells and macrophage-like RAW 264.7 cells. Thus, in combination, TILI-2 demonstrates a strong potential to be used in targeting hyperpigmentation disorders. Our LC-MS/MS studies showed that TILI-2 might also target the TGF-β signaling pathway when reducing melanin production. This was corroborated by real-time PCR evaluation of specific genes.

## Figures and Tables

**Figure 1 molecules-27-03228-f001:**
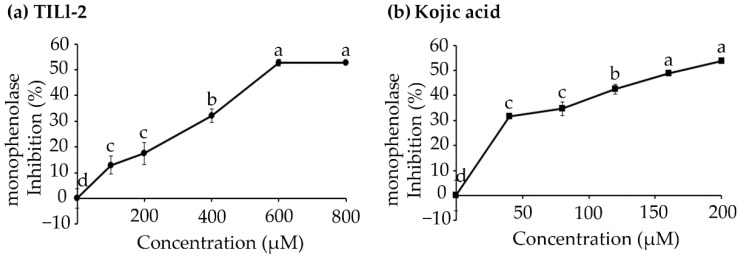
The effect of (**a**) TILI-2 peptide and (**b**) kojic acid on monophenolase inhibitory activity. TILI-2 was incubated with tyrosinase and L-tyrosine at various concentrations from 0 to 800 µM. As a control, kojic acid was used at concentration from 0 to 200 µM. Monophenolase inhibition (%) = (1 − A/B) × 100, where A and B are the initial rates (∆A475/min) of the enzyme reaction with and without the peptide inhibitor, respectively. All values represent the mean ± SD. Different letters indicate the significant differences from Duncan’s test (*p* < 0.05).

**Figure 2 molecules-27-03228-f002:**
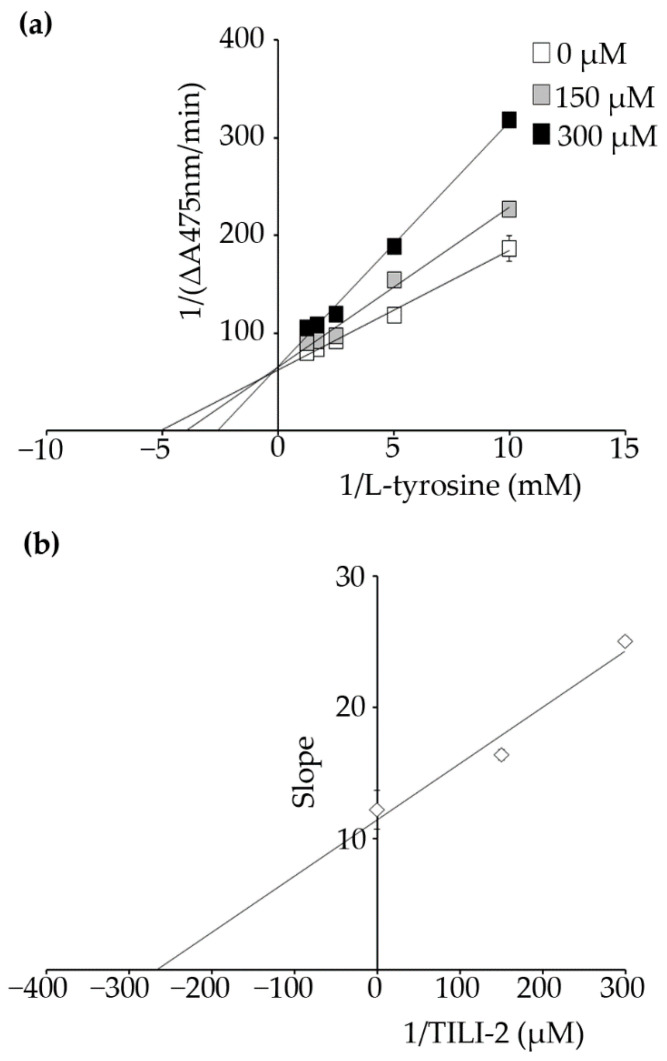
Kinetic study of TILI-2 reveals it is a competitive inhibitor. (**a**) Lineweaver-Burk plots with TILI-2 at a range of concentrations from 0 to 300 µM. (**b**) Secondary plot of Lineweaver-Burk plot. *K*_i_ was obtained from the plot of slope versus TILI-2 concentration.

**Figure 3 molecules-27-03228-f003:**
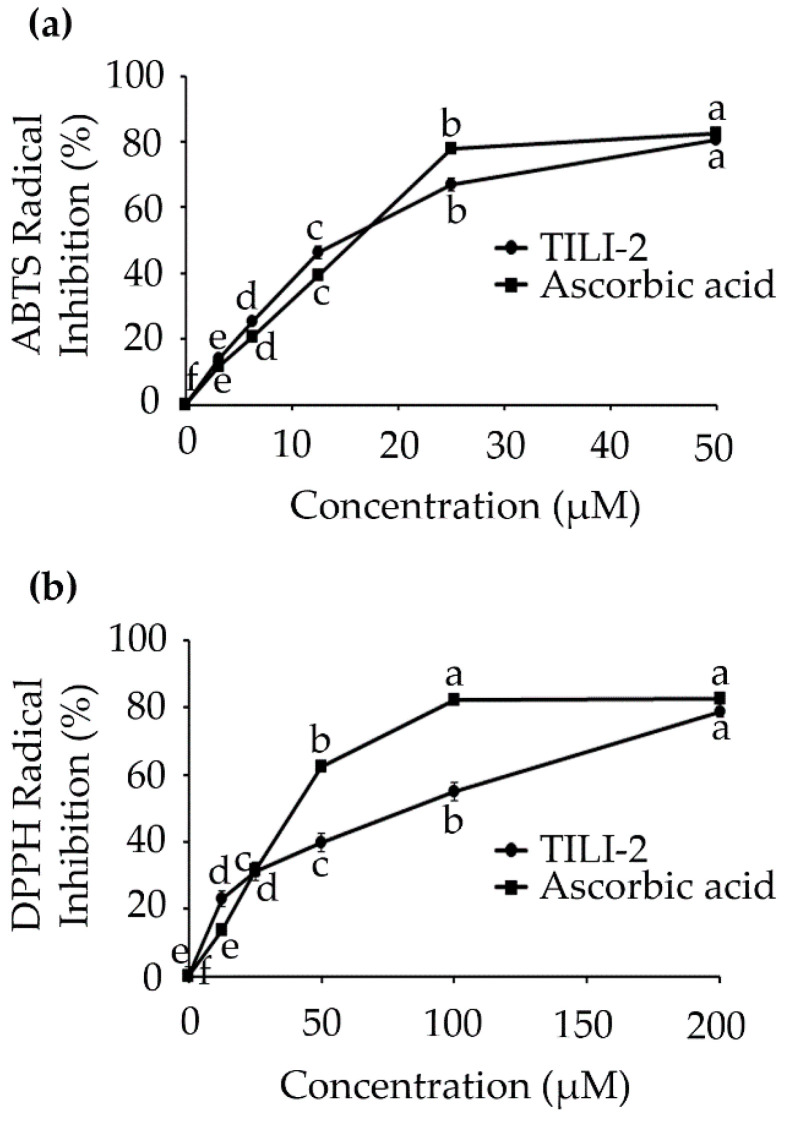
TILI-2 displays an antioxidant activity. (**a**) ABTS and (**b**) DPPH radical scavenging activities of TILI-2 and positive control ascorbic acid at concentrations ranging from 0 to 200 µM for DPPH radical and 0 to 50 µM for ABTS radical. All values represent the mean ± SD. Different letters indicate the significant differences from Duncan’s test (*p* < 0.05).

**Figure 4 molecules-27-03228-f004:**
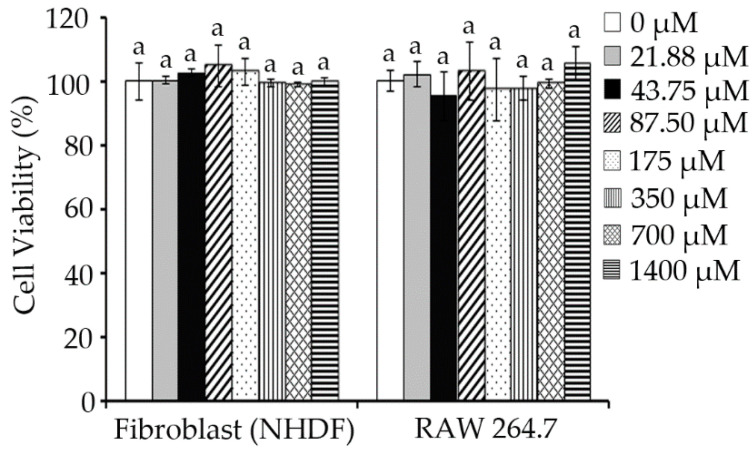
TILI-2 displays negligible cellular cytotoxicity. The cell viability of fibroblast (NHDF) cells and RAW 264.7 cells upon treatment with various concentrations of TILI-2 (0–1400 µM) is shown. All values represent the mean ± SD. Different letters indicate the significant differences from Duncan’s test (*p* < 0.05).

**Figure 5 molecules-27-03228-f005:**
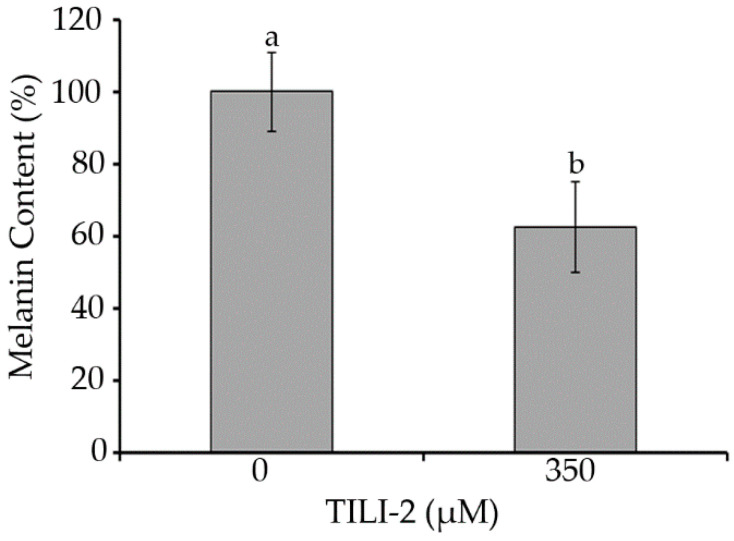
TILI-2 reduces melanin content. B16F1 cells were treated with TILI-2 (350 µM). All values represent the mean ± SD. Different letters indicate the significant differences from Duncan’s test (*p* < 0.05).

**Figure 6 molecules-27-03228-f006:**
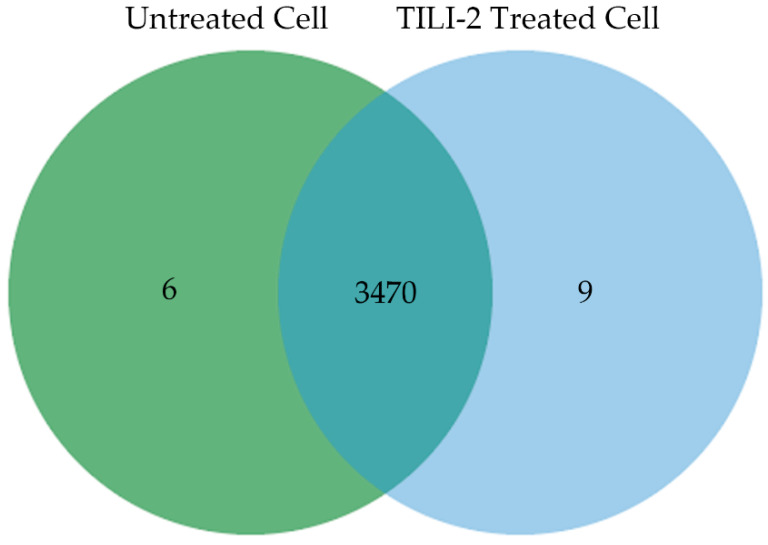
Venn diagram of identified proteins of untreated B16F1 cells overlapped with TILI-2 treated B16F1 cells.

**Figure 7 molecules-27-03228-f007:**
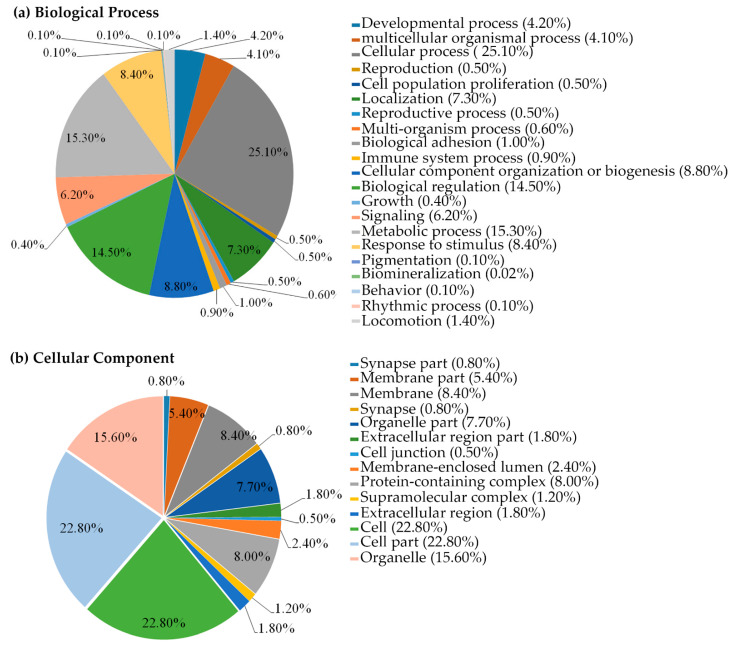
Gene-ontology (GO) functional annotation of biological processes and cellular components. The gene-ontology (GO) functional annotation of 3485 expressed proteins in TILI-2 untreated and treated cells were classified using the PANTHER classification system according to (**a**) biological processes and (**b**) cellular components.

**Figure 8 molecules-27-03228-f008:**
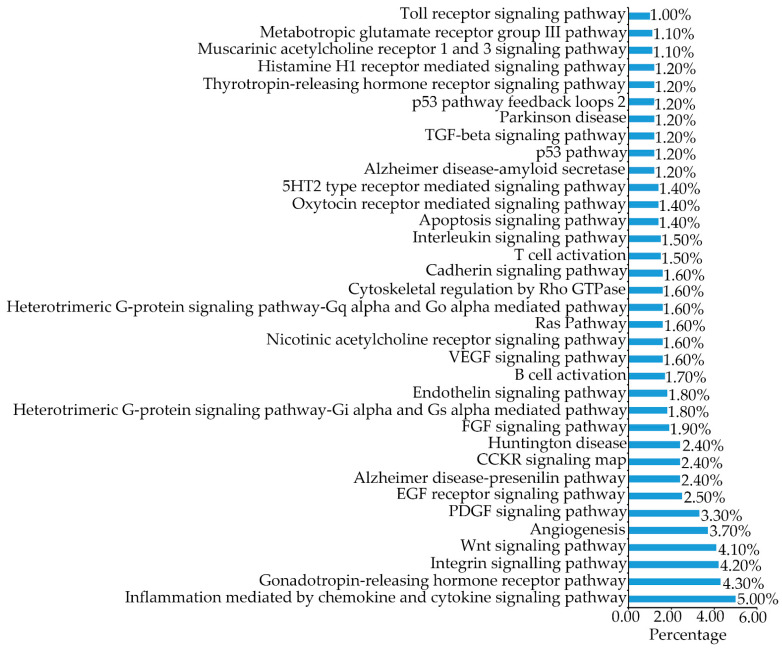
Gene-ontology (GO) functional annotation for cell signaling pathways. The gene-ontology (GO) functional annotation of 3485 expressed proteins in TILI-2 untreated and treated cells were classified using the PANTHER classification system according to the cell signaling pathway.

**Figure 9 molecules-27-03228-f009:**
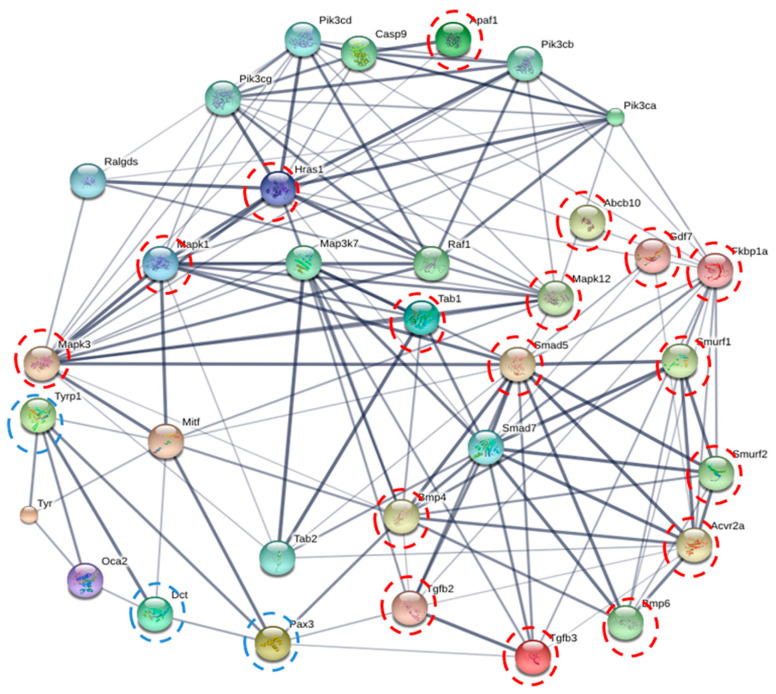
Protein–protein interactions by STITCH. The interactive networks of 17 proteins in the TGF-β signaling pathway and their predicted association with proteins in melanogenesis. The red dashed line circles indicate the proteins in the TGF-β pathway, which directly interact with the melanogenesis proteins. The blue dashed line circles depict the proteins in melanogenesis from the proteomics dataset.

**Figure 10 molecules-27-03228-f010:**
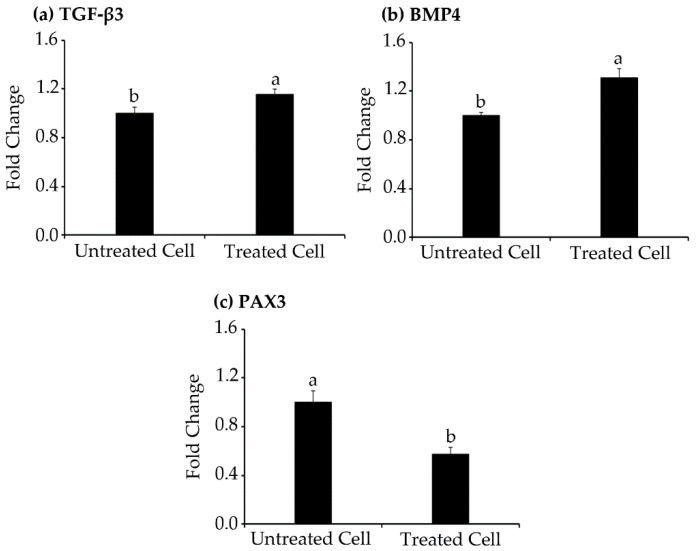
Gene expression. The effect of TILI-2 (350 µM) on the cellular mRNA for TGF-β3 (**a**), BMP4 (**b**), and PAX3 (**c**) in B16F1 cells. All values represent the mean ± SD. Different letters indicate the significant differences from Duncan’s test (*p* < 0.05).

**Table 1 molecules-27-03228-t001:** The half-maximal monophenolase inhibitory concentration (IC_50_) of TILI-2 against monophenolase activity of tyrosinase.

Peptide Name	Sequence	IC_50_ (µM)
TILI-2	CNGVQPK	582.92 ± 6.44 ^a^
Kojic acid	-	173.20 ± 6.44 ^b^

Mean ± SD was used to represent the values. Different letters indicate the significant differences from Duncan’s test (*p* < 0.05).

**Table 2 molecules-27-03228-t002:** Effect of TILI-2 on kinetic constants of tyrosinase monophenolase activity.

Name	Concentration (µM)	*K*_m_ or *K’*_m_ (µM)	*V*_max_ or *V’*_max_ (△A475/min)	Mode of Inhibition	*K*_i_ (µM)
TILI-2	0	196.58 ± 36.64 ^a^	0.016 ± 0.0010 ^a^	competitive	273.39 ± 72.90
	150.00	250.09 ± 10.65 ^b^	0.015 ± 0.0003 ^a^		
	300.00	381.30 ± 10.77 ^c^	0.015 ± 0.0002 ^a^		

Mean ± SD was used to represent the values. Different letters indicate the significant differences from Duncan’s test (*p* < 0.05).

**Table 3 molecules-27-03228-t003:** The half-maximal inhibitory concentration (IC_50_) of TILI-2 against ABTS and DPPH radicals.

Peptide Name	Sequence	IC_50_ (µM)	
		ABTS Radical	DPPH Radical
TILI-2	CNGVQPK	16.78 ± 0.48 ^a^	89.83 ± 2.72 ^a^
Ascorbic acid	-	15.86 ± 0.16 ^a^	40.09 ± 0.21 ^b^

Mean ± SD was used to represent the values. Different letters indicate the significant differences from Duncan’s test (*p* < 0.05).

**Table 4 molecules-27-03228-t004:** The relative quantitation ratios (log2) of proteins in the TGF beta signaling pathway and proteins involved in melanogenesis upon treatment with TILI-2 from proteomics data.

Function	Order	Protein Name	Gene Name	*t*-Test *(p)*	Average Log2 Expression Value	
					Untreated Cell	TILI-2 Treated Cell
TGF-beta pathway	1	Mitogen-activated protein kinase 1	MK01_MOUSE	0.9510	14.99	15.01
	2	Mitogen-activated protein kinase kinase kinase 7-interacting protein 1	TAB1_MOUSE	0.9878	15.62	15.63
	3	Mothers against decapentaplegic homolog 5	SMAD5_MOUSE	0.7259	15.18	15.26
	4	Transforming growth factor beta-3	TGFB3_MOUSE	0.8527	14.79	14.86
	5	GTPase HRas	RASH_MOUSE	0.3821	14.07	14.25
	6	Mitogen-activated protein kinase 3	MK03_MOUSE	0.6866	14.79	14.61
	7	E3 ubiquitin-protein ligase SMURF1	SMUF1_MOUSE	0.5889	12.76	14.39
	8	E3 ubiquitin-protein ligase SMURF2	SMUF2_MOUSE	0.0265	15.33	15.58
	9	Growth/differentiation factor 5	ABCBA_MOUSE	0.3343	16.03	16.39
	10	Bone morphogenetic protein 6	BMP6_MOUSE	0.3300	15.96	16.51
	11	Growth/differentiation factor 7	GDF7_MOUSE	0.5095	15.48	13.72
	12	Bone morphogenetic protein 4	BMP4_MOUSE	0.6398	15.86	16.00
	13	Activin receptor type-2A	AVR2A_MOUSE	0.6189	15.52	15.71
	14	Apoptotic protease-activating factor 1	APAF_MOUSE	0.9578	14.13	13.99
	15	Mitogen-activated protein kinase 12	MK12_MOUSE	0.5613	13.22	13.55
	16	Peptidyl-prolyl cis-trans isomerase FKBP1A	FKB1A_MOUSE	0.1541	14.53	13.37
	17	Transforming growth factor beta-2	TGFB2_MOUSE	0.0009	15.47	16.10
Melanogenesis	18	L-dopachrome tautomerase	TYRP2_MOUSE	0.1481	14.26	14.65
	19	5,6-dihydroxyindole-2-carboxylic acid oxidase	TYRP1_MOUSE	0.4880	16.09	16.28
	20	Paired box protein Pax-3	PAX3_MOUSE	0.3759	13.35	11.55

**Table 5 molecules-27-03228-t005:** Primer sequences for real-time PCR.

Gene Name	Primer Sequence (5’→3’)	Annealing Temperature (°C)	Product Size (bp)	Reference Sequence
TGF-β3	F_TCTCTGTCCACTTGCACCAC	57	137	NM_009368.3
	R_TGATAGGGGACGTGGGTCAT			
BMP4	F_AGGGATCTTTACCGGCTCCA	57	111	NM_007554.3
	R_ACTCCTCACAGTGTTGGCTC			
PAX3	F_ACATCTCAGCCCTATTGTCCC	57	105	NM_008781.4
	R_CGTCCAAGGCTTACTTTGTCC			
GAPDH	F_CGTCCCGTAGACAAAATGGT	57	110	NM_008084.3 [62,63]
	R_TTGATGGCAACAATCTCCAC			

## Data Availability

Not applicable.

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
