# Peer review of "Evaluation of TILI-2 as an Anti-Tyrosinase, Anti-Oxidative Agent and Its Role in Preventing Melanogenesis Using a Proteomics Approach"

_molecules, 2022, doi:10.3390/molecules27103228_

Round 1
Reviewer 1 Report
This is a well-written manuscript reporting TILI-2 as a potential depigmenting agent with adequate evidence from a properly designed research study. Just a few minor suggestions are given below:
- The term 'toxicity' used in the manuscript can be replaced with cytotoxicity.
- Line 308: Km and Vmax need to be defined.
Author Response
Response to reviewers for manuscript molecules-1707667 entitled
Evaluation of TILI-2 as an Anti-Tyrosinase, Anti-Oxidative Agent and Its Role in Preventing Melanogenesis Using a Proteomics Approach
We greatly appreciate the reviewers’ time and helpful comments, which have enabled us to improve this manuscript. Please see our responses to the reviewers’ comments below:
Reviewer 1:
Comment: The term 'toxicity' used in the manuscript can be replaced with cytotoxicity
Response: We have replaced the term “toxicity” with cytotoxicity in line 27, 69, 166, 167, 170, 179 and 532.
Comment: Line 308: Km and Vmax need to be defined
Response: The Km and Vmax definitions have been added to the manuscript in line 356-360 as shown below:
… inhibitory activity assay (section 2.3). Vmax, the rate of the reaction when enzyme is fully saturated, and Km, the Michaelis constant or the concentration of substrate at which the enzyme achieves half of its maximum rate of reaction, were obtained from a Lineweaver-Burk plot. These values were used to specify the mode of inhibition.

Reviewer 2 Report
Joompang et al. evaluate the capabilities of TILI-2 on anti-tyrosinase, anti-oxidative using various biochemical assays as well as elaborate the potential role of this peptide in preventing melanogenesis using proteomics technique. The manuscript is generally well-written. Prior to acceptance of this manuscript for publication, I would suggest the following revisions:
- Line 52, please introduce the full name of TILI-2 since it shows firstly in this manuscript.
- Section 1, Line 58: Please correct the typo, TILI-1, to TILI-2.
- Section 2.1, Figure 1, and section 3.3: The authors used IC50, which is a measurement of the potency of a substance in inhibiting a specific function (i.e. monophenolase in this section), to describe the inhibitory activity of TILI-2 in section 2.1.; while they used monophenolase activity (%), which means the activity of the enzyme itself, on Figure 1 to illustrate the inhibitory activity of TILI-2 to monophenolase. This is very confusing. Same issue found in method section 3.3., the authors named the section monophenolase inhibitory activity assay, but the description of this section was about the determination of the monophenolase activity. Please correct them and provide the equation of how to calculate inhibitory activity (%).
- Section 2.3, Figure 3, and section 3.5&3.6: The authors used IC50 to describe the antioxidant activity of TILI-2 in section 2.3.; while they used the measurement of still active ABTS/DPPH radical (%) on Figure 3 to illustrate the antioxidant activity of TILI-2. This is very confusing. Same issue found in method section 3.5&3.6. Radical scavenging (%) activity was expressed as the inhibition concentration (IC50), however, the equation which the authors used was for the determination of the left free radical (%), not the inhibited free radical (%). Please correct them and provide the equation of how to calculate radical scavenging (%).
- Figure 1: The bar chart, which presents the inhibitory activity of both TILI-2 and Kojic acid, is not a common way on presenting the dose-dependent inhibitory activity of investigating compound. Please present the monophenolase inhibitory activities of TILI-2 and the positive control, kojic acid, in a line chart.
- Figure 3: The bar chart, which present the antioxidant activities of TILI-2 and Ascorbic acid to ABTS+ and DPPH+, is not a common way on present the does-dependent antioxidant activity of investigating compound. Please present the results in line charts.
- Method 3.8: In this section, the authors only did 24-hour incubation after peptide was added. Did the authors do any pilot study to learn about the potential time-dependent manner?
Author Response
On behalf of my co-authors and myself, I am enclosing a revised version of manuscript molecules-1707667 entitled Evaluation of TILI-2 as an Anti-Tyrosinase, Anti-Oxidative Agent and Its Role in Preventing Melanogenesis Using a Proteomics Approach. for possible publication in the Molecules. We greatly appreciate and acknowledge the valuable comments raised by the referee and we have revised the manuscript according to the suggestions.

Reviewer 3 Report
In the manuscript entitled "Evaluation of TILI-2 as an Anti-Tyrosinase, Anti-Oxidative Agent and Its Role in Preventing Melanogenesis Using a Proteomics Approach", the authors evaluate peptide derivative TILI-2 towards prevention of melanogenesis by active as an anti-tyrosinase and anti-oxidative agent. Authors have used appropriate research design and used appropriate techniques and the manuscript is comprehensive.
I recommend authors provide, in the introduction, some relevant references on currently available alternatives in prevention of melanogenesis and what would be the benefit of TILI-2 and similar peptide-based approach, in general. This would make the rational of the study clear to a reader.
Author Response
Response to reviewers for manuscript molecules-1707667 entitled
Evaluation of TILI-2 as an Anti-Tyrosinase, Anti-Oxidative Agent and Its Role in Preventing Melanogenesis Using a Proteomics Approach
We greatly appreciate the reviewers’ time and helpful comments, which have enabled us to improve this manuscript. Please see our responses to the reviewers’ comments below:
Reviewer 3:
Comment: I recommend authors provide, in the introduction, some relevant references on currently available alternatives in prevention of melanogenesis and what would be the benefit of TILI-2 and similar peptide-based approach, in general. This would make the rational of the study clear to a reader.
Response: As per the reviewer’s suggestion, we have added the following information to the introduction:
Bioactive peptides are a promising group of depigmenting agents, because they are derived from natural sources and therefore display low cytotoxicity. For example, the peptides produced by hydrolysis of the jellyfish proteins have displayed good antioxidative and tyrosinase inhibitory activity [13], while purified jellyfish collagen peptide (JCP) has shown melanogenesis-inhibitory activity [14,15]. CT-2 (LQPSHY), a C-terminus tyrosine peptide derived from rice bran, even repressed melanogenesis in mouse B16 melanomas [16].
- Upata, M.; Siriwoharn, T.; Makkhun, S.; Yarnpakdee, S.; Regenstein, J.M.; Wangtueai, S. Tyrosinase Inhibitory and Antioxidant Activity of Enzymatic Protein Hydrolysate from Jellyfish (Lobonema smithii). Foods 2022, 11, doi:10.3390/foods11040615.
- Zhuang, Y. Antioxidant and melanogenesis-inhibitory activities of collagen peptide from jellyfish (Rhopilema esculentum). Journal of the science of food and agriculture 2009, v. 89, pp. 1722-1727-2009 v.1789 no.1710, doi:10.1002/jsfa.3645.
- Ab Aziz, N.A.; Salim, N.; Zarei, M.; Saari, N.; Yusoff, F.M. Extraction, anti-tyrosinase, and antioxidant activities of the collagen hydrolysate derived from Rhopilema hispidum. Prep Biochem Biotechnol 2021, 51, 44-53, doi:10.1080/10826068.2020.1789991.
- Ochiai, A.; Tanaka, S.; Imai, Y.; Yoshida, H.; Kanaoka, T.; Tanaka, T.; Taniguchi, M. New tyrosinase inhibitory decapeptide: Molecular insights into the role of tyrosine residues. J Biosci Bioeng 2016, 121, 607-613, doi:10.1016/j.jbiosc.2015.10.010.

This manuscript is a resubmission of an earlier submission. The following is a list of the peer review reports and author responses from that submission.